# Using Standstill Time to Evaluate the Startup in Polymer Pair Systems

**DOI:** 10.3390/polym15244696

**Published:** 2023-12-13

**Authors:** Anita Ptak, Zuzanna Łuksza

**Affiliations:** Department of Fundamentals of Machine Design and Mechatronic Systems, Faculty of Mechanical Engineering, Wroclaw University of Science and Technology, ul. I. Lukasiewicza 5, 50-371 Wroclaw, Poland; zuzanna.luksza@gmail.com

**Keywords:** standstill time, sliding pair, unit pressure, polymer–polymer, static friction coefficient, friction

## Abstract

The subject of polymer–polymer pair interaction is highly important, bearing in mind that such pairs are used in the construction of machines and equipment, among other uses. Considering that the characteristics of polymer–polymer sliding pairs (e.g., the load limit value and advantageous parameter, PV) differ from those of polymer–metal pairs, the subject is particularly interesting and has been little explored so far. Hence, the present study presents one of the areas of the effects of standstill time (intrinsically characteristic of polymeric materials) on the startup parameters in sliding pairs where the sample and the countersample were made of a polymeric material. Pairs of same-type polymers, POM–POM, PET–PET, and PA6–PA6, were subjected to tests. A test rig dedicated to static friction coefficient determination, whose principle of operation is based on the interdependences between the force characteristics of an inclined plane, was used for this purpose. The sliding pair was successively loaded with 25 N, 50 N, and 75 N, and the standstill time ranged from 0 to 10 min. The determined tribological characteristics were analysed with regard to the standstill time under load, unit pressure, and polymer pair material. An optical profilometer and a scanning electron microscope were used to qualitatively evaluate the effects of standstill time and unit pressure on the surfaces of the interacting elements. Complex interrelationships between the test results and the set experimental parameters were noted. SEM micrographs revealed post-friction changes in the sliding surfaces.

## 1. Introduction

Polymer tribology, as a field of science dealing with the investigation of the interactions between contacting surfaces, is vitally important today as polymers are increasingly often used in various engineering applications. The understanding and control of the startup in polymer pair systems is crucial in many fields, such as the polymer industry, the automotive industry, nanotechnology, biomedicine, and so on. The phenomena and the processes taking place during startup initiation have a bearing on the reliability and durability of the combination. The static friction coefficient is a parameter used to characterise the instant of a startup. Static friction occurs when there is no noticeable macroscopic relative displacement between bodies. According to Coulomb’s law [1], static friction is proportional to the normal load, where the coefficient of friction is defined as a factor of proportionality. At the macroscopic scale, one can observe micromotions initiated by the external forces acting on a system. A reduction in frictional resistance means a lower energy expenditure needed to initiate a relative motion, and so, energy and operating cost savings. Static friction is also a key factor having a bearing on the stability of form–fit connections. Without a proper frictional resistance, such connections can be susceptible to shifting, which can result in structural damage.

In the literature on the subject, one can find tribological investigations of sliding pairs made up of different materials, e.g., steel–steel, steel–ceramic, and ceramic–ceramic pairs [2,3,4,5,6]. However, owing to the advantageous properties of polymeric materials, a metal–polymer pair is most often used in the construction of machines and equipment. There are many studies devoted to the evaluation of the properties of such sliding pairs in a general context [7,8,9,10,11] and for specific uses, e.g., in bearings [12,13], gears [14], and medical devices [15]. One can also find tribological investigations of not only conventionally produced materials (by extrusion), but also 3D printed materials [16], investigations conducted under different conditions such as during dry friction and using various lubricating media [17], and studies in which various kinds of motion (e.g., rotational motion, alternating motion, micromotion, etc.) are used [18]. In addition, there are variables connected with material properties. These variable parameters assumed in tests result in countless possibilities. Moreover, there is one more material pair—a polymer–polymer pair—for which there are much fewer tribological studies than for the other cases. The interaction between polymeric materials attracted the attention of Erhard [19] and Maeda [20], who unanimously ascribed the phenomena occurring at the interface between two polymers to the work of adhesion, which is proportional to the force of static friction. Jia et al. [21] found the friction and wear properties of a polymer–polymer combination to be closely dependent on the PV-value for a sliding pair working under dry friction conditions, whereas the PV-value was found to have little effect on the tribology of behaviour in the case of oil lubrication. Rymuza et al. highlighted the significant role of the starting of polymer–polymer trunnion microbearings [22,23,24]. According to their results, the time needed for slip to occur increases with static contact duration prior to shear stress initiation. It is believed that this effect is due to an increase in preliminary displacement, i.e., the length of the elongation of the material until its failure, characteristic of the transition from static contact to sliding. Rymuza ascribes the increase in preliminary contact surface area to a change in the modulus of elasticity caused by the rheological deformation of the contact. Another interesting phenomenon observed during the interaction between polymers is the transfer of material from one polymer onto another polymer. Bahadur in [25] showed that spontaneously formed transfer film can be modified by introducing certain composite additions and thereby its tribological behaviour can be modelled. Some fillers affect the development of the transfer film and increase its adhesion to the substrate. Such fillers reduce the rate of wear of a polymer. On the other hand, there are many fillers that have no effect on the transfer film and in such cases, the wear is increased.

The interrelationships described above are bound with surface roughness, which is considered to be one of the key parameters being a point of reference for friction process investigations. In their book [26], Bowden and Tabor have already shown that the contact between materials at rest is a sum of microcontacts which form due to the existing surface roughness and which are randomly distributed [27]. The force needed to break this contact is directly linked with the energy accumulated on the surface of the microcontacts [26], and since the latter are bound with local roughness, the friction coefficient is thought to have locally a random value [28]. An additional obstacle (even at the microscale) to reducing randomness is surface texturing, whose ultimate purpose is to improve tribological properties [15,29]. One should note, however, that surface roughness is not the only factor having a bearing on the friction process. The latter also depends on the unit pressure, the relative speed of the interacting materials, the temperature, the humidity, and the environment in which the tests are conducted. Therefore, when conducting tribological tests and analysing their results, care must be taken to perform proper statistical calculations.

Most often, an undesirable effect of static friction is stick–slip, i.e., uncontrolled jerky movements of the body being moved. In the case of sliding pairs, it is essential that sliding start smoothly, whereas stick–slip results in undesirable consequences, such as energy losses, surface damage (wear), failures of subassemblies, or simply noise generation. Hu et al. in [2] showed that slip initiation depends on the contact between the surfaces characterised by roughness. When the surfaces are characterised by high roughness, the contact is deep. The initiation of motion in this case involves a change in the dislocation plasticity bound with strong mechanical interactions on the rubbing surfaces. In the case of smooth surfaces, a shallow contact occurs and the force needed to break it depends on adhesion; the latter is determined by the duration of the standstill under load [30]. In the literature, it is assumed that the static friction value depends on the duration of the static contact between materials [3,31]. Filippov et al. [32] corroborated that the probability of the formation of a bond at the interface between roughness apexes depends on the duration of the contact, and so, a shorter time available for bonding during sliding can explain some of the differences between the force needed to start motion and the force maintaining the motion. Li et al. in [33] proved that an increase in a static friction value with a static load is bound with the development of chemical bonds on roughness apexes. Furthermore, recent research has shown that much prior to macroscopic slips, significant contact morphology modifications, including both a reduction in the actual contact surface and an increase in anisotropy, take place. It is thought that these changes, caused by shear, potentially affect all the macroscopic properties of a sliding pair under static load [27,34,35]. In the above context, one should consider polymeric materials, especially the ones belonging to the group of thermoplastics, the peculiar properties of which affect the friction process. During a tribological analysis, one should take into account the mechanical response of viscoelastic materials. Considering the surface imperfections (roughness and undulations), the loading history, and the rheology of viscoelastic materials, it is difficult to determine relevant supportive mathematical relations on the basis of the existing analytical theories. Therefore, the solution of the above questions is sought not only through fundamental laboratory research, but also through numerical and theoretical studies. In this way, the randomness of local friction properties can be combined with the observed variation at the macroscopic scale, contact and friction models can be developed, and the changes in contact morphology caused by shear can be taken into account [5,28,36]. A relevant robust numerical model was developed by Chen et al. [37]. It proposes a solution to the problem of the point contact between a rigid indenter and a homogenous viscoelastic surface. This model in simulations takes into account polymeric materials distinguished by a wide range of relaxation time and a complicated surface topography. Shortly afterward, the adhesion phenomenon was incorporated into the model [38].

It has been found that because of the complexity of the process, the reported experimental results can differ even when an attempt is made to faithfully reproduce the test procedure. Even though the differences are not frequently reported, they are an important factor considering that there is no complete and cohesive theory of friction [19,20]. An analysis of both static friction and the changes taking place at the interface of interacting polymeric materials during the initiation of motion are key to assuring the durability of sliding junctions in many applications. For this reason, the present paper is devoted to a tribological analysis of selected polymeric materials: POM–POM, PET–PET, and PA6–PA6. The tribological analysis is based on the determination of the change in the coefficient of static friction in relation to the standstill time under static load (*t*) and the set unit pressure (*p*). The tribological tests were supplemented with a qualitative evaluation of the conditions of the rubbing surfaces in order to identify the processes and phenomena taking place during the interaction.

## 2. Materials and Methods

### 2.1. Samples

Three sliding pairs of same-type polymers, POM–POM, PA6–PA6, and PET–PET (Table 1), were subjected to tests. The choice of these thermoplastic materials was dictated by their wide use, ready availability, and outstanding sliding properties (low friction coefficient). An additional consideration for the choice of a material was to keep the variety of materials to a minimum, which is one of the basic principles of pro-recycling design. The tribological tests were carried out using the pin-on-plate setup. Hence, the samples were prepared in two shapes: 10 × 100 × 5 mm plates and a Ø8 × 15 mm pin (Figure 1).

### 2.2. Method of Measuring Friction

The static friction coefficient of the polymer–polymer sliding pairs was investigated on a pin-on-plate test rig (Figure 2). The test rig was equipped with a tilting arm, a rotary pan, an electric actuator, a laser indicator, weights, and a sample fixture. More weights could be put on the rotary pan in order to investigate the friction coefficient under various unit pressures. The design of the pan, via whose rotary system a force was applied to the friction junction, ensured that regardless of the angle of inclination, the vector of the force stemming from the additional load was applied at the sample–countersample interface level. The electric actuator drove the tilting arm.

For the test, the polymer sample was mounted on the horizontally positioned tilting arm, while the polymeric pin was set up on the countersample and loaded by the rotary pan and weights. Prior to each measurement, the rubbing surfaces were cleaned and degreased. The electric actuator attached to one end of the tilting arm made it possible to slowly and uniformly change the arm’s angle of inclination, and so, the inclination of the sliding pair. At the instant of static contact breaking, i.e., when the pin was sliding on the plate, an inclination reading was taken. Formula (1) was used to calculate the static friction coefficient and the actual pressure force could be determined from Equation (2):(1)$$\mu_{0}=\tan\rho =\frac{H}{L}=\frac{F_{T}}{F_{N}}$$
where $\mu_{0}$—static friction coefficient, ρ—friction angle, *H*—the inclination height of the sliding pair, *L*—distance of the tilting arm rotation axis from the measurement point, *F_T_*—friction force, and *F_N_*—normal force.
(2)$$F_{N}=F_{G}\cdot \cos\rho$$
where *F_N_*—normal force, *F_G_*—gravitational force, and ρ—friction angle.

Three polymer–polymer sliding pairs, POM–POM, PA6–PA6, and PET–PET, were subjected to the test. Each of the pairs was tested under three loads: 25 N, 50 N, and 75 N. Weights were mounted on the pan whose deadweight amounted to 940 g. The weights together with the pan corresponded to unit pressures of, respectively, 0.65 MPa, 1.1 MPa, and 1.6 MPa. The static load times for each of the pairs amounted to, respectively, 0 min, 2.5 min, 5 min, 7.5 min, and 10 min. The measurements were performed 10 times. The basic test parameters are presented in Table 2.

The static load time was selected on the basis of previous studies [6,39,40,41,42]. A load time ranging from 1 min to over 100 h was adopted. Since the greatest changes were observed in the first 10 min of contact, attention was focused on the analysis of tribological properties around the instant of startup.

### 2.3. Evaluation of Rubbing Surface Conditions

Regardless of the materials used, as two elements rub against each other, macro-/microscopic interactions occur on the rubbing surfaces. Many factors can have a bearing on these changes. In tribology, these are mainly mechanical interactions arising as a result of the action of forces in the contact zone, thermal changes due to the change in friction energy, and chemical reactions causing all kinds of tribological wear [43]. Therefore, the surfaces of the interacting materials (the pin and the plate) were subjected to microscopic examinations. A Leica DCM8 3D Surface Metrology Microscope (Leica Microsystems, Wetzlar, Germany) was used to evaluate the surface structure. Owing to its two-core system, this microscope combines the advantages of confocal microscopy and interferometry. The measurements were performed in the confocal mode, characterised by the highest measuring accuracy in tests of polymeric materials, and the entire measurement was contactless. Moreover, the condition of the sliding surfaces was imaged using a Phenom-World ProX electron microscope (ThermoFisher Scientific, Waltham, MA, USA). Thanks to the use of this technique, the processes that took place during friction were qualitatively imaged.

## 3. Results and Discussion

### 3.1. Analysis of the Effects of Standstill Time and Unit Pressure on the Static Friction Coefficient

The results of the tribological tests carried out on the pairs of same-type polymers are contained in Table 3. The relationships between the static friction coefficient and the set parameters, determined on the tribological rig, are presented in Figure 3. The test results were analysed with regard to standstill time, unit pressure, and the polymer pair used. The tests were repeated 10 times for each configuration of set parameters, whereby the statistical calculations customarily used in tribological investigations could be conducted. The calculated confidence interval was used to statistically evaluate the results as regards the probability of committing an error. A confidence level of α = 5% was assumed to calculate confidence limits. The whole was subjected to Dixon’s test to eliminate gross errors. The results are graphically presented in the form of error bar diagrams.

The determined characteristics showed that in each of the tested pairs, the highest unit pressure-induced static friction coefficient values occurred at a load time of t = 2.5 min. In the case of a unit pressure of *p* = 1.1 MPa, the highest static friction coefficient characterised the POM–POM polymer pair, followed by the PET–PET pair, while the lowest coefficient characterised the PA6–PA6 pair. It is interesting to note that this relationship did not hold for the load time at t = 10 min. The lowest unit pressure-induced static friction coefficient values were observed for the longest load time.

At a medium unit pressure of *p* = 0.65 MPa, the static friction coefficient value was higher for POM–POM and increased with load time to μ_av_.. ≈ 0.35, whereas for pairs PET–PET and PA6–PA6, it decreased to μ_av_. ≈0.31 (PET–PET) and μ_av_.. ≈ 0.20 (PA6–PA6). At the unit pressure of *p* = 1.1 MPa, the static friction coefficient depending on the load time at rest increased for PA6–PA6 and PET–PET, whereas it decreased for POM–POM. At the unit pressure of *p* = 1.6 MPa, the static friction coefficient decreased with load time at rest in the case of each pair. It decreases fastest for PET–PET (from μ_av_.. ≈ 0.35 to μ_av_. ≈ 0.29) and PA6–PA6 (from μ_av_.. ≈ 0.21 to μ_av_. ≈ 0.18).

### 3.2. Qualitative Analysis of the Rubbing Surfaces

Friction surface topography was examined using an interferometric profilometer using a non-contact measurement method. This work focused on presenting surface maps and surface axonometrics. Additionally, surface roughness was determined using the S index as the parameter adopted in the three-dimensional analysis. It should be noted that all the polymer samples were prepared using the same machining parameters.

Figure 4, Figure 5 and Figure 6 show the 2D view (surface map) and 3D view (surface axonometric) of the rubbing surfaces of the polymeric materials. The values of the surface roughness parameters of the measured surface areas are compiled in Table 4.

After the tribological tests, the surfaces of the elements that had rubbed against each other showed similar parameter values. The roughness of the pins was found to be lower than that of the plates. In each of the cases, the pin surface parameters have lower values than those of the plate. Exemplary photographs of the surface topography for the particular surfaces show friction marks in the form of surface asperities. This could be one of the causes of the relatively high value of parameter Sa.

### 3.3. Evaluation of the Phenomena Taking Place on the Rubbing Surfaces

The macroscopic examinations were carried out using a scanning electron microscope. A backscattered electron detector working in the topographic mode was used during the examinations. Microscopic images were obtained at the accelerating voltage of 15 kV. Selected SEM micrographs of the sliding surfaces of the polymeric materials are shown in Figure 7, Figure 8 and Figure 9.

The SEM micrographs show changes in the sliding surfaces of the tested materials. Friction marks of various orientations are observed on the surfaces of both the pin and the plate. Distinct grooves running along the path of friction, exhibiting relatively stronger directionality than in the case of the other surfaces, are visible in Figure 7. The grooves are very deep as indicated by the surface profile obtained using an optical profilometer and the roughness value. The SEM micrographs show wear products on each of the tested surfaces, whose presence is evidence of their participation in the friction process (Figure 7 and Figure 8). Figure 8 shows a situation in which a wear product, once again taking part in friction, found itself in the contact zone and was pressed onto the sliding surface. The irregular shape and the wavy edges are evidence of the high plasticity of this wear product. Also, deep and wide grooves, especially in Figure 9a, are visible. PET is characterised by the smoothest post-friction face with a small number of scratches (Figure 8). The small, but long wavy edges (similarly to the wear product pressed onto the surface) could be indicative of the high plasticity of at least the material’s surface itself.

### 3.4. Summary of Results

Tribological tests were conducted for three sliding pairs of same-type polymers: POM–POM, PET–PET, and PA6–PA6. Unit pressure and standstill time under static load were the variable parameters in the experiment. After the test cycle was run, the rubbing surfaces were subjected to a topographic analysis to determine the conditions of the surfaces of the tested materials and to evaluate the phenomena occurring during friction. Depending on the unit surface pressure, differences between the values of the static friction coefficients of the tested sliding pairs of same-type polymers are noticeable (Figure 10). The causes of the variation could be ascribed to the differences in both the strength and elastic properties of the materials used.

The highest friction coefficient values were noted for the POM–POM pair (0.29–0.37). The relatively high values can be due to the condition of the material surface after the test. Numerous surface defects (large pits—valleys/cavities) are visible on both the pin and the plate (Figure 4 and Figure 7). The surface irregularities are reflected in the determined roughness parameters. Parameter Sku, used to evaluate surface defects, took the value of above 3 for almost all the surfaces, which means there was an increase in the number and/or size of peaks/valleys on the surface. Furthermore, negative Ssk parameter values indicate a surface characterised by deep cavities/valleys, i.e., a “plateau surface” [44]. This can contribute to greater mechanical friction, intensifying friction resistances. Such pits/valleys in the surface can be beneficial, as lubricant-collecting places, for lubricated joints. In the case of technically dry friction, they can be places where wear products collect.

The lowest static friction coefficient value is noted for the PA6–PA6 pair (0.18–0.23). This is interesting considering that PA6 when interacting with steel is characterised by a relatively high friction coefficient [26,45]. It is probable that this is connected with the polyamide’s creep compliance [46,47]. During steady static loading, the deformation of the surface changes due to the relocation of chains in the surface layer [47]. Such changes can lead to a reduction in the resistance to motion. Furthermore, the surface micrographs show a certain plasticity of the layer (Figure 9), manifesting itself in undulated edges of the friction marks. Because of the short time of the actual interaction, a change in the temperature at the contact between the interacting surfaces was rather ruled out here.

For each of the tested sliding pairs, the graph of the static friction coefficient at a load of *p* = 1.1 MPa and the longest load time at rest of t = 10 min shows a different tendency than at the other loads. At a static load time of t < 10 min, a distinct upward or downward trend is visible for the tested polymer pairs. At t = 10 min and a unit pressure of *p* = 1.1 MPa, a deviation—a considerable change in the friction coefficient—is noticeable. It is hard to seek explanations here, especially because of reports to the contrary in the literature [11,28]. To settle this issue, the range of the investigation was extended as regards static load time.

The PA6–PA6 sliding pair showed the most stable characteristic (in terms of both the value of the static friction coefficient and accuracy). As this material is characterised by a lower hardness [48], the interactions at the contact between the polymers were bound to have advantageous ratios of elastic-to-plastic interactions. The other two tested materials are characterised by a higher hardness and stiffness, whereby the irregularities on the surface of the samples can act as microblades, increasing the mechanical component of the friction force.

The static friction coefficient values without a load time were similar, whereas at longer standstill times and a change in unit pressure, the values diverged. (Figure 3). This divergence could be due to various causes, e.g., to the influence of viscoelasticity, i.e., variable time-dependent strains and stresses. The longer it takes to reach a static load, the higher the probability that creep, i.e., a phenomenon which appears after a system imbalanced by the action of external forces regains its balance, will occur. What is critical here is the time needed for a deformation to develop during which the elements of the polymer’s structure must displace. This has a direct bearing on the shape of the sample and its properties [49]. The friction coefficient at the unit pressure of 2.5 MPa and that at the unit pressure of 5 MPa differed by just under 5%, i.e., within the assumed error. A similar situation was observed for the PET–PET pair, where at the standstill times of 2.5 min, 5 min, and 7.5 min, the difference was even smaller, amounting to less than 3%.

It is interesting to note that for the heaviest loading of the sample, the friction coefficient values were lowest or at least equal to those for the other two loads. This applied to the whole range of standstill times under loads, where t > 0. The area contact between two bodies occurs as a result of the deformation of the peaks of surface irregularities, whose sum total of areas constitutes the actual area of the friction contact. Under longer loading, the surface layer undergoes transformation, whereby the actual contact zone increases. This enables the formation of adhesive bonds, which at the molecular level, constitute an additional force increasing resistance before motion initiation.

## 4. Conclusions

On the basis of the tests and the comprehensive tribological characteristics obtained in this way, the conclusions drawn consist of the following:The test results clearly show that changing the process parameters for the tested polymer–polymer sliding pairs significantly impacts both the unit pressure and the standstill time under static load on the startup parameters of the friction system. A general downward tendency is visible for the static friction coefficient depending on the load time at rest.For the particular loads, an increase in stationary contact time affects the static friction coefficient value, but it does not reveal unequivocal relationships between the materials. As regards sliding properties, the PA6–PA6 pair has the most advantageous tribological characteristics. The relatively low coefficient of static friction was due to an increase in the actual contact area and a decrease in the actual unit pressure.A marked abrupt change in the static friction coefficient value for the longest adopted load time of t = 10 min was visible (at a downward tendency it began to rise and vice versa). The literature reports quite different findings on the static friction coefficient, but investigated for the metal–polymer pair [11,28]. According to reports, the greatest tribological changes take place during motion initiation after a very short load time—up to a maximum of t = 10 min. At longer times, these characteristics are already more stable and constant. Hence, one can conclude that the general characteristics ascribed to various relationships have not been corroborated for the tested polymer–polymer sliding pairs.On the basis of the identified scratches, grooves, and wear products, one can conclude that, unlike the elastic response, the transition from the partial to full slip of the viscoelastic materials exhibits a dynamic tendency. When identical contact parameters are determined for different viscoelastic materials, one can expect a quicker transition from the partial slip to the full slip at the different microhardness and thermal conductivity of the tested materials [28,50].

The input data (*p*, t) used in these investigations are only a few parameters having a bearing on the course of the friction process. In order to gain a full picture and to fully explain the complexity of the processes and phenomena accompanying the interaction between polymers, further research into this subject is necessary. To follow up the results obtained as part of this study, it would be advisable to extend the tests to cover a longer standstill time under load and add a lubricant.

## Figures and Tables

**Figure 1 polymers-15-04696-f001:**
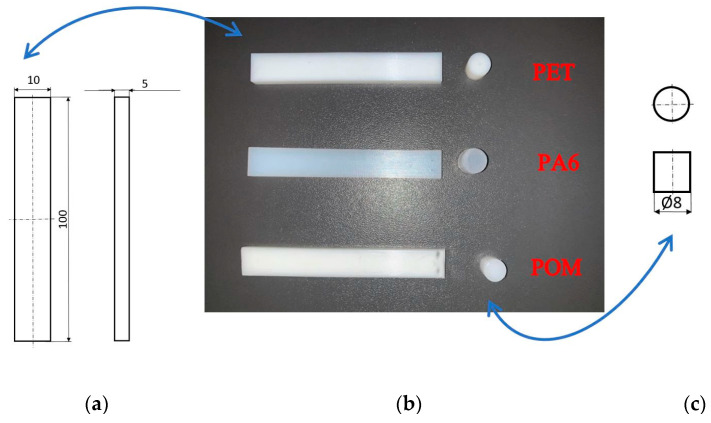
Samples of tested materials: (**a**) diagram of plate, (**b**) photos of samples of tested materials, (**c**) diagram of pin.

**Figure 2 polymers-15-04696-f002:**
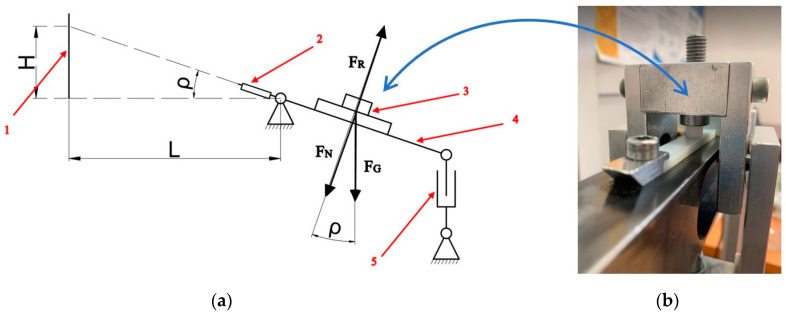
Test rig for measuring static friction coefficients. (**a**) Test rig diagram: 1—graduated strip; 2—laser indicator; 3—sliding pair; 4—tilting arm; 5—electrical actuator; H—inclination height of the sliding pair; L—distance of the tilting arm rotation axis from the measurement point; ρ—friction angle; F_N_—normal force; F_G_—gravitational force; F_R_—reactive force. (**b**) Photo of sliding pair.

**Figure 3 polymers-15-04696-f003:**
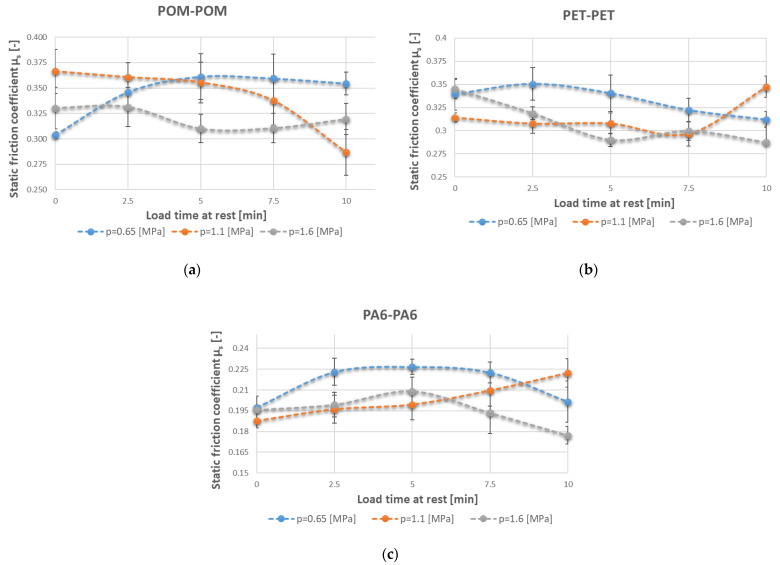
Static friction coefficients versus loading times at rest for sliding pairs (**a**) POM–POM, (**b**) PET–PET, and (**c**) PA6–PA6, for a whole cycle of unit pressures.

**Figure 4 polymers-15-04696-f004:**
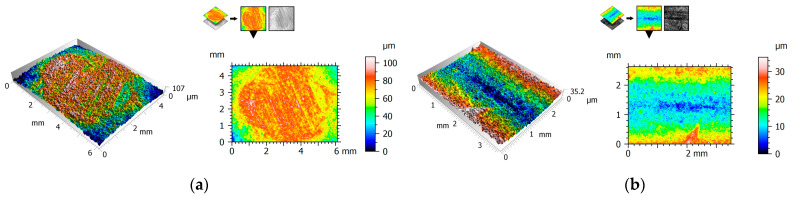
2D and 3D views of POM: (**a**) pin, (**b**) plate.

**Figure 5 polymers-15-04696-f005:**
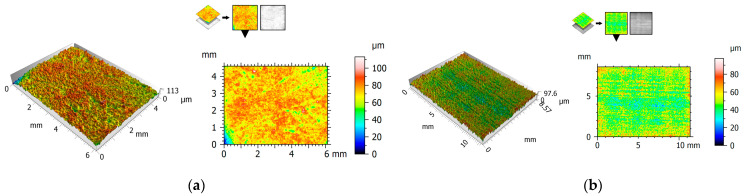
2D and 3D views of PET: (**a**) pin, (**b**) plate.

**Figure 6 polymers-15-04696-f006:**
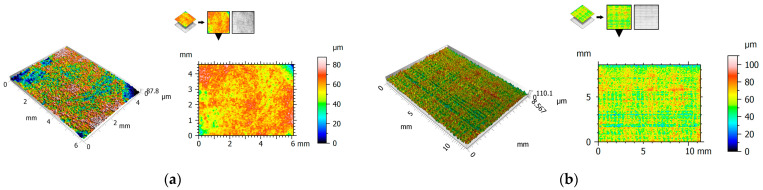
2D and 3D views of PA6: (**a**) pin, (**b**) plate.

**Figure 7 polymers-15-04696-f007:**
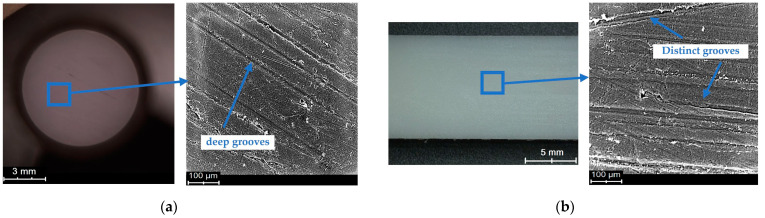
SEM and optical micrograph of POM sliding surface: (**a**) pin, (**b**) plate.

**Figure 8 polymers-15-04696-f008:**
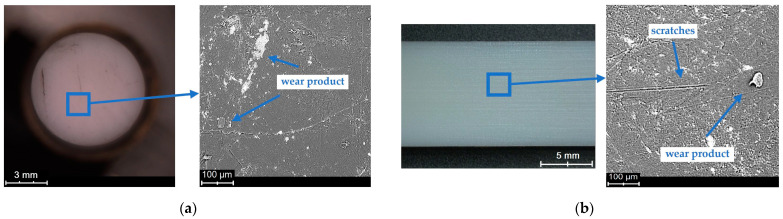
SEM and optical micrograph of PET sliding surface: (**a**) pin, (**b**) plate.

**Figure 9 polymers-15-04696-f009:**
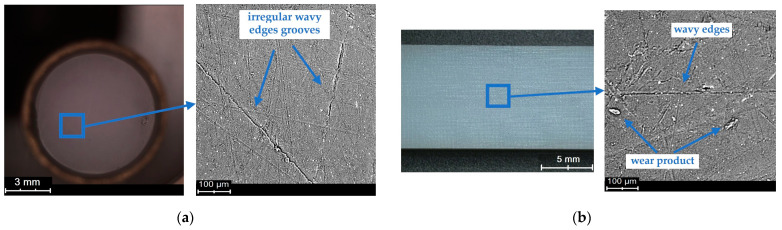
SEM and optical micrograph of PA6: (**a**) pin, (**b**) plate.

**Figure 10 polymers-15-04696-f010:**
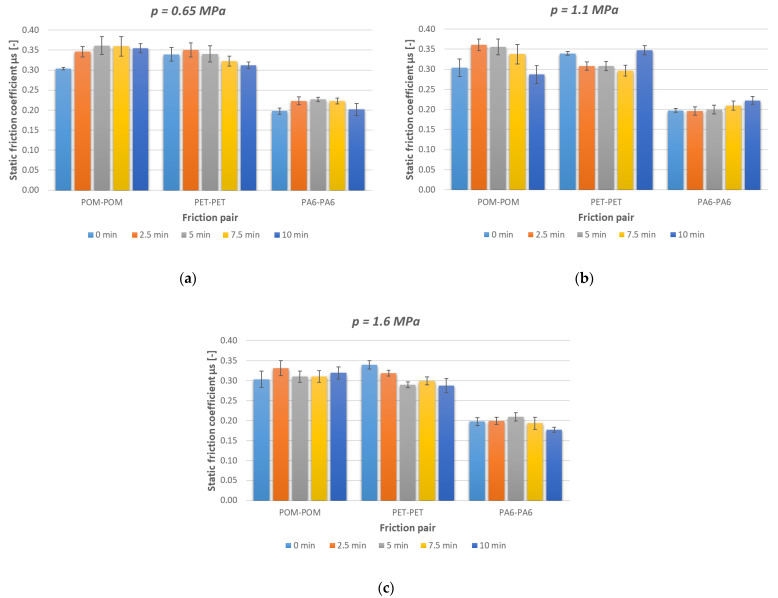
Static friction coefficients versus friction pairs for (**a**) *p* = 0.65 MPa, (**b**) *p* = 1.1 MPa, and (**c**) *p* = 1.6 MPa, for a whole cycle of loads at rest.

**Table 1 polymers-15-04696-t001:** Selected properties of tested polymeric materials.

Property	POM	PA6	PET
Composition	Non-modified	Non-modified	Non-modified
Yield point, R_e_	62 MPa	60 MPa	80 MPa
Modulus of tensile elasticity, E_r_	2.7 GPa	1.80 GPa	2.8 GPa
Long-term use temperature	100 °C	100 °C	110 °C
Glass transition temperature, *T_g_*	−60 °C	+60 °C	70 °C
Density, ρ	1.41 g/cm^3^	1.13 g/cm^3^	1.37 g/cm^3^
Water absorption, W_s_	0.5%	9.5%	0.5%

**Table 2 polymers-15-04696-t002:** Tribological test parameters.

Experimental Parameters	
Polymer sliding pair pin-on-plate	POM–POM
	PA6–PA6
	PET–PET
Weight of pan	~9.40 N
Load force	25 N, 50 N, 75 N
Standstill time	0–10 min
Environment	Technically dry friction
Number of repetitions	10

**Table 3 polymers-15-04696-t003:** Average values of static friction coefficients for pairs of same-type polymers.

Standstill	POM–POM						PET–PET						PA6–PA6					
Time	0.65 MPa		1.1 MPa		1.6 MPa		0.65 MPa		1.1 MPa		1.6 MPa		0.65 MPa		1.1 MPa		1.6 MPa	
(min)	µ_av_.	α	µ_av_.	α	µ_av_.	α	µ_av_.	α	µ_av_.	α	µ_av_.	α	µ_av_.	α	µ_av_.	α	µ_av_.	α
0	0.30	0.00	0.37	0.02	0.33	0.02	0.34	0.02	0.31	0.00	0.35	0.01	0.20	0.01	0.19	0.01	0.20	0.01
2.5	0.35	0.01	0.36	0.01	0.33	0.02	0.35	0.02	0.31	0.01	0.32	0.01	0.22	0.01	0.20	0.01	0.20	0.01
5	0.36	0.02	0.36	0.02	0.31	0.01	0.34	0.02	0.31	0.01	0.29	0.01	0.23	0.01	0.20	0.01	0.21	0.01
7.5	0.36	0.02	0.34	0.02	0.31	0.01	0.32	0.01	0.30	0.01	0.30	0.01	0.22	0.01	0.21	0.01	0.19	0.02
10	0.35	0.01	0.29	0.02	0.32	0.02	0.31	0.01	0.35	0.01	0.29	0.02	0.20	0.01	0.22	0.01	0.18	0.01

**Table 4 polymers-15-04696-t004:** Roughness parameters of rubbing surfaces of tested materials.

Roughness Parameters	POM		PET		PA6	
	Pin	Plate	Pin	Plate	Pin	Plate
Sq (µm)	5.725	13.411	8.069	9.948	8.430	11.965
Ssk (–)	0.421	−0.879	−0.301	0.041	−0.849	−0.299
Sku (–)	2.444	4.005	3.512	3.114	5.879	3.300
Sp (µm)	20.306	37.743	39.548	50.159	29.652	51.929
Sv (µm)	14.919	69.155	41.054	47.472	58.114	58.186
Sz (µm)	35.225	106.898	80.602	97.631	87.766	110.115
Sa (µm)	4.806	10.487	6.322	7.928	6.338	9.441

## Data Availability

The data presented in this study are available upon request from the corresponding author.

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
