# Peer review of "Using Standstill Time to Evaluate the Startup in Polymer Pair Systems"

_polymers, 2023, doi:10.3390/polym15244696_

Round 1
Reviewer 1 Report
Comments and Suggestions for Authors
Remarks to the paper
Using standstill time to evaluate the startup in polymer pair systems
I. Main remarks
1) paper bases on a very reach examination of materials (tribological, profilometric, SEM) and is worth to be published, but need same additive information and explanation of results;
2) Figures have to small numbers by description and are difficult to read
3) The paper need a deeper analyze and discussion of the results. Presented in chapter 3. Results and Discussion and in 3.4. Summary of results are almost a verbal description of the results given in tables and I diagrams. The reader of scientific journal can see it.
For example page 10 lines 290-291 “.. The highest friction coefficient values were noted for POM-POM pair (0.29-0.37) and the lowest for the PA6-PA6 pair (0.18-0.23). What is the reason for it?
Page 11 lines 296-298 What was happened in the contact area during a longer standstill time?
Please try to answer why, by using of achieved results, micrographs roughness profiles, materials properties chains in polymers, etcetera.
Please try to give answers for questions Why are the achieved result so? What is the reason for registered levels and facts.
4. Conclusions should be rewritten according to presented remarks.
Conclusion 3 “Thanks to SEM examination it was possible to qualitatively evaluate the polymeric sliding surface”
For this information have Authors examined polymer contacts?
II. Detailed remarks
1) Pleas add description of symbols used in Fig. 2 and equations 1 and 2;
2) “The topography of the surfaces was examined by means of an interferometric profilometer, describing not only the graphical surface topography, but also determining the surface roughness of the tested materials.”
This sentence is not clear, because roughness parameters are a part of a surface topography.
3) Figs. 4-5
Diameter of 3D profile is 6 mm, but the diameter of pin is 8 mm. Haw was the 3D profiles measured?
In pictures 4-5 are not presented “photograps”, but the 3D-roughness profiles. In some software is used such term “photographs”.
What would Authors show on this pictures?
4) Table 4
Units of high roughness parameters are needed (µm?).
Haw was the inaccuracy of roughness measuring?
Given parameters are for example Sa=4.80579, what suggests an accuracy of 0.01 nanometer.
Please use an appropriate numbers rounding.
Why Authors have measured roughness parameters given in Tab. 4, for example Sq, Ssk, since has not used to explain the results.
5) Figs-7-9
Description numbers of microphotographs are too small, and difficult to read. Please enlarge it.
Please add to the optical photographs places from which are the SEM micrograph made. This allows to show the picture position to the movement direction. Some scratches on the surfaces after rubbing are compatible with this direction.
6) Page 10 line 270
“The groves are very deep as indicated by the surface profile obtained usin an optical profilometer and the roughness value”
Was this depth measured?
It would be good to add a 2D roughness profile throw this grooves and measure is deptht.
7) Line 274
“… wear product …was pressed onto the sliding surface.”
What is an evidence for this statement?
Haw was prepared the samples after tribological tests, before SEM examining? Was it ultrasonic cleaned?

Author Response
Dear Reviewer.
Thank you very much for your feedback and the opportunity to refer to it in this response. The authors appreciate any suggestion to preserve the quality and completeness of the submitted manuscript.
Please see the attached file. You will find the answer to the review there.
Best regards

Reviewer 2 Report
Comments and Suggestions for Authors
The manuscript 'using standstill time to evaluate the startup in polymer pair systems' reported th static friction of three pairs of polymers, under different loading conditions. This work is within the scopes of Polymers, and the manuscript is well organised.
To further improve the manuscript, the authors are advised to consider the following:
1) line 153, please confirm the dimension of the plate used.
10 x 1000 x 5 mm (reported in line 153), or 100 x1000 x5 mm (reported in Fig 1)
2) height of the pin is not reported.
3) line 176, please define FT.
4) line 181: 5 N or 50 N?
5) (3.2 qualitative analysis of the rubbing surfaces)
Fig 6 showed the profile of speciment after the sliding test.
should the surface profile before the test be presented also? surface roughness of the sliding pairs before the sliding test are important to know in this work.
6) Surface roughness parameters,- are the results obtained from the average results from 10 tests?
7) magnification scales are not readable on Figs 7-9.
8) arrows were added on Figs 7-9. please add the information you wishes to highlight on the figures.
9) please further explain line 303 "the test results show that as the load time at rest increases, some predictability of the static friction coefficient value is lost".
10) Unsure why p =0.6 MPa is named as 'medium' unit pressure.
11) if the objective of this research is to compare the static friction of three different polymers, then a static friction graph with 3 polymers should be presented.
Author Response
Dear Reviewer.
Thank you very much for your feedback and the opportunity to refer to it in this response. The authors appreciate any suggestion to preserve the quality and completeness of the submitted manuscript.
Please see the attached file. You will find the answer to the review there.
Best regards.

Reviewer 3 Report
Comments and Suggestions for Authors
tribological properties of the polymer-polymer pair are discussed in the paper for some materials selected by the authors. There is a good introduction, a clear abstract and well-written text. But the paper does not have at least any analysis done, namely: why for one material static friction is like this, and for another it is like that. No attempt has been made to explain. Results and conclusion do not have useful information that allow us to understand what is the matter and why one polymer gives such friction, and another polymer gives another, and why they have different relationships over time. It is necessary to add physical analysis to the Discussion section, to connect friction with the properties of materials and their surfaces in order to get out of the category of a very narrow case study.
Author Response

(The authors gave the same response as above.)

Round 2
Reviewer 2 Report
Comments and Suggestions for Authors
The authors have revised the paper and addressed most of the previous comments.
If the objective of this research is to compare the static friction of three different polymers, then a static friction graph with 3 polymers should be presented.
Reviewer 3 Report
Comments and Suggestions for Authors
The revised version of the paper can be published